# Coronary Microvascular Dysfunction in Acute Cholestasis-Induced Liver Injury

**DOI:** 10.3390/biomedicines12040876

**Published:** 2024-04-16

**Authors:** Sebastian Billig, Marc Hein, Celine Kirchner, David Schumacher, Moriz Aljoscha Habigt, Mare Mechelinck, Dieter Fuchs, Uwe Klinge, Alexander Theißen, Christian Beckers, Christian Bleilevens, Rafael Kramann, Moritz Uhlig

**Affiliations:** 1Department of Anesthesiology, Faculty of Medicine, RWTH Aachen University, Pauwelsstraße 30, 52074 Aachen, Germanycbleilevens@ukaachen.de (C.B.);; 2Department of Nephrology and Clinical Immunology, Faculty of Medicine, RWTH Aachen University, Pauwelsstraße 30, 52074 Aachen, Germany; 3FUJIFILM VisualSonics, Inc., Joop Geesinkweg 140, 1114 AB Amsterdam, The Netherlands; 4Department of General, Visceral and Transplantation Surgery, Faculty of Medicine, RWTH Aachen University, Pauwelsstraße 30, 52074 Aachen, Germany

**Keywords:** acute cholestatic liver injury, cardiac function, liver–heart axis, rodent model, coronary microvascular dysfunction, myocardial contrast echocardiography, coronary flow velocity reserve

## Abstract

Background: Previous studies have shown cardiac abnormalities in acute liver injury, suggesting a potential role in the associated high mortality. Methods: We designed an experimental study exploring the short-term effects of acute cholestasis-induced liver injury on cardiac function and structure in a rodent bile duct ligation (BDL) model to elucidate the potential interplay. Thirty-seven male *Sprague-Dawley* rats were subjected to BDL surgery (*n* = 28) or served as sham-operated (*n* = 9) controls. Transthoracic echocardiography, Doppler evaluation of the left anterior descending coronary artery, and myocardial contrast echocardiography were performed at rest and during adenosine and dobutamine stress 5 days after BDL. Immunohistochemical staining of myocardial tissue samples for hypoxia and inflammation as well as serum analysis were performed. Results: BDL animals exhibited acute liver injury with elevated transaminases, bilirubin, and total circulating bile acids (TBA) 5 days after BDL (TBA control: 0.81 ± 2.54 µmol/L vs. BDL: 127.52 ± 57.03 µmol/L; *p* < 0.001). Concurrently, cardiac function was significantly impaired, characterized by reduced cardiac output (CO) and global longitudinal strain (GLS) in the echocardiography at rest and under pharmacological stress (CO rest control: 120.6 ± 24.3 mL/min vs. BDL 102.5 ± 16.6 mL/min, *p* = 0.041; GLS rest control: −24.05 ± 3.8% vs. BDL: −18.5 ± 5.1%, *p* = 0.01). Myocardial perfusion analysis revealed a reduced myocardial blood flow at rest and a decreased coronary flow velocity reserve (CFVR) under dobutamine stress in the BDL animals (CFVR control: 2.1 ± 0.6 vs. BDL: 1.7 ± 0.5 *p* = 0.047). Immunofluorescence staining indicated myocardial hypoxia and increased neutrophil infiltration. Conclusions: In summary, acute cholestasis-induced liver injury can lead to impaired cardiac function mediated by coronary microvascular dysfunction, suggesting that major adverse cardiac events may contribute to the mortality of acute liver failure. This may be due to endothelial dysfunction and direct bile acid signaling.

## 1. Introduction

Acute liver failure (ALF) is defined as a coagulopathy (international normalized ratio > 1.5) with a concomitant hepatic encephalopathy without a preexisting liver disease [1,2]. The underlying severe hepatocellular injury can be due to a broad range of triggers. Drug toxicity (e.g., paracetamol) and infectious diseases such as hepatitis are two common factors. The incidence in Western industrialized countries is around 5.5 cases per million population per year. Although this is comparatively low, it is still a life-threatening disease with a mortality rate at around 40% [3]. Until now, cerebral edema, hemorrhage, and sepsis, but less frequently cardiac complications, have been thought to be responsible [4]. However, clinical studies increasingly suggest a relevant cardiac involvement. As early as 1976, Weston et al. demonstrated that more than 90% of the ALF patients they studied had concomitant cardiac abnormalities, of which 25% experienced sudden cardiac death [5]. Additionally, a study by Parekh et al. showed that 74% of ALF patients had elevated cardiac troponin I (cTnI) levels correlating with increased morbidity and mortality. Thus, it is conceivable that subclinical myocardial injury plays a critical role in ALF [6,7] or that elevated cTnI levels are at least a reflection of metabolic stress caused by coronary microvascular dysfunction (CMD) for instance [8]. The term refers to a spectrum of structural and functional disorders of the coronary microcirculation, which affect the coronary blood flow (CBF) and can ultimately result in myocardial ischemia [9]. Myocardial blood flow is usually matched to oxygen demand in the coronary vasculature by coordinating resistance within different microvascular regions (conductive arteries, pre-arterioles, and arterioles), which are subject to distinct regulatory mechanisms [10]. Disruption of these mechanisms is called CMD and may result in an unmet oxygen demand. It can occur in the context of both obstructive and non-obstructive coronary artery disease. Alternatively, it may occur as a functional disorder in the absence of underlying primary heart disease [9].

Building upon these observations, our previous research has shown that major adverse cardiac events (MACE) are a critical determinant of mortality in ALF patients [11]. Furthermore, we hypothesized that inflammation, endothelial dysfunction, and direct bile acid signaling may be involved in the pathophysiology of secondary myocardial injury in ALF as we observed a dropout rate of 20% early after BDL in a previous study, some even with macroscopically visible infarcts [11].

The rodent BDL model is a widely used animal model reflecting acute and chronic cholestasis-induced live injury depending on the duration of the experiment. Ligation of the common bile duct results in marked cholestasis and inflammation. Within the first seven days after BDL, acute cholestatic live injury occurs, followed by chronic injury with increasing fibrosis of the liver [12]. The BDL model is therefore the most widely used model to induce obstructive cholestatic injury in rodents and to study the molecular and cellular events underlying these pathophysiological mechanisms [12,13,14,15].

Therefore, we designed a 5-day rodent BDL study to further elucidate the short-term effect of acute cholestatic live injury on myocardial function, structure, and perfusion and to test the hypothesis that acute cholestatic liver injury leads to a reduced cardiac function with an impaired myocardial microcirculation and subsequent inflammation. The aim of our study is to further elucidate the contribution of cardiac complications to the mortality of acute liver injury and, thus, to gain insights that will enable us to develop strategies to reduce them.

## 2. Materials and Methods

### 2.1. Animals and Housing

The experimental procedures were performed on 37 male *Sprague-Dawley* rats (Rattus norvegicus; RjHan:SD; Janvier Labs, Le Genest Saint Isle, France) weighing 500 ± 23 g on average. The animals were housed in rat filter cages (type 2000, Tecniplast, Hohenpreisenberg, Germany) with a 12 h light–dark cycle under specific pathogen-free conditions in a temperature- and humidity-controlled environment.

Prior to any experimental procedures, the animals were acclimatized for at least 7 days. The rats had free access to sterile, acidified water, and standard rat laboratory chow (Sniff GmbH, Soest, Germany) throughout the experiment. All experimental procedures adhered to the German Animal Welfare Act (§ 8 Abs. 1, Tierschutzgesetz [16]) and were approved by the governmental animal care and use office (No 84-02.04.2016. A391, Landesamt für Natur, Umwelt und Verbraucherschutz Nordrhein-Westfalen, Recklinghausen, Germany). The manuscript reporting corresponds to the ARRIVE guidelines 2.0 [17].

During the entire study, the animals were visited at least once a day to assess the general condition of the rats using a semiquantitative score sheet. Body weight, general condition, spontaneous behavior, and behavioral criteria were considered in the assessment (5–9 points = low stress, 10–19 points = moderate stress, and ≥20 points = high stress) [18]. In cases where animals had more than 20 points or 10–19 points for more than 72 h, animal welfare officers were consulted immediately, and animals were euthanized immediately if necessary.

### 2.2. Bile Duct Ligation Surgery and Echocardiography

On day 0, echocardiography was conducted following analgesia induced by subcutaneous administration of buprenorphine (0.01–0.03 mg/kg body weight (BW), Temgesic, Essex Pharma GmbH-Msd Sharp & Dohme GmbH, Haar, Germany) 30 min before the procedure. Anesthesia was induced by 4 vol% isoflurane and maintained by 2 vol% in 100% oxygen applied via nosecone to the spontaneously breathing rats. Animals were placed on a rat handling platform (FUJIFILM VisualSonics, Toronto, ON, Canada) with integrated heating pad in a supine position. The ventral areas of the thorax and abdomen were shaved. Transthoracic echocardiography (TTE) was performed using a Vevo 3100 imaging system (FUJIFILM VisualSonics) with an MX 250 linear array transducer (FUJIFILM VisualSonics). Animals were allowed to rest for 5 min before imaging to achieve stable and comparable hemodynamic conditions. Three B-mode loops of the parasternal long axis (PLAX) were recorded for further analysis. The left coronary artery (LCA) was identified with color Doppler in a modified PLAX and then examined with Pulsed-wave Doppler (PWD) as described by Billig et al. [19].

After TTE rats were transferred to a feedback loop-controlled heating pad (TCAT-2LV controller, Physitemp, Clifton, NJ, USA) in supine position. A ligation of the common bile duct (CBD) was performed, as described by Tag et al. [15]. Adequate depth of anesthesia was ensured by checking interdigital reflexes. The abdominal cavity was opened via a median laparotomy, followed by dissection of the CBD. It was ligated twice with a silk suture 5/0 (18020-50, Fine Science Tools, Vancouver, BC, Canada) and separated between the ligations. Toward the end of surgery, the wound margin was infiltrated with ropivacaine (Ropivacaine Kabi 10 mg/mL, Fresenius Kabi AG, Kriens, Switzerland) (2.5 mg/kg BW). In addition, dipyrone (100 mg/kg BW diluted to 100 mg/mL; novaminsulfon ratio 1 g/2 mL, Ratiopharm, Ulm, Germany) was administered subcutaneously 30 min prior to the end of anesthesia and additionally on the first postoperative day. Sham-operated animals served as controls. These animals received the same treatment without ligation and transection of the CBD. After recovery from anesthesia, animals were transferred to their cages; 28 animals were treated by BDL, and 9 animals received a sham operation. The group allocation was quasi-randomized. Blinding of the investigators was not possible as the BDL rats were visually recognizable by their jaundice.

### 2.3. Final Surgery and Echocardiography

Five days post-BDL, animals underwent a final examination. Anesthesia was performed as described above. After the ventral part of the neck had been shaved, a double-lumen central venous catheter (Multicath-2, 3Fr, Vygon, Ecquen, France) was placed in the left external jugular vein using a cutdown technique. In addition to the PLAX images and Doppler assessment of the LCA, myocardial contrast echocardiography (MCE) was performed on the 5th postoperative day. MCE was performed using nonlinear contrast mode in a PLAX view of the left ventricle. A systolic gating (R-wave triggered) was activated during MCE. Images were acquired as described previously in Billig et al. [19]. MCE measurements were performed in triplicate. Noninvasive blood pressure measurements were performed using an inflatable tail cuff with a pulse sensor (IN125/R, ADInstruments Ltd., Oxford, UK). Blood pressure, TTE, Doppler, and MCE imaging were performed under baseline conditions as described above as well as under intravenous infusion of 140 µg/kg BW/min adenosine, and intravenous infusion of 10 µg/kg BW/min dobutamine. To ensure maximum drug effect, images were acquired five minutes after the start of the continuous infusion.

After completion of the echocardiographic examination, blood samples were withdrawn via the central venous catheter, and animals were euthanized under deep anesthesia. Finally, organs were harvested to obtain histological specimens.

### 2.4. Analysis of Echocardiographic Examination

Echocardiographic images were analyzed using Vevo Lab 5.6 (FUJIFILM VisualSonics). End-diastolic volume (VED), stroke volume (SV), cardiac output (CO), and ejection fraction (EF) were measured using the auto-LV function. Coronary diastolic peak velocity (DPV), systolic peak velocity (SPV), and velocity time integral (VTI) were determined from the PWD signal. Coronary flow velocity reserve (CFVR) was determined for adenosine and dobutamine as the ratio of DPV during dobutamine or adenosine, consecutively, and DPV at baseline [20]. All data presented were averaged from three consecutive cardiac cycles. The MCE loops were truncated to a length of 45 s after the contrast agent arrived. A ventricular region of interest (ROI) was placed in the ventricular cavity and a myocardial ROI in the anteroseptal wall of the left ventricle and time-intensity curves (TIC) for both ROIs were generated [19]. Thus, peak enhancement (PE), wash-in rate (WIR), and area under the curve (AUC) were taken from the TICs. Measurements were averaged over three consecutive bolus applications and normalized to the ventricular ROI for each animal and then referred to as normalized values [21]. Rate pressure product (RPP) was determined as the product of the heart frequency and systolic blood pressure.

### 2.5. Histological Staining

All tissue samples were fixed in 4% buffered formalin (ROTI Histofix 4%, Carl Roth GmbH + Co. KG, Karlsruhe, Germany) for one week, then embedded in paraffin and cut into 3 µm sections. Slides prepared were then dewaxed and rehydrated using a standard xylol and a descending alcohol series. 

#### 2.5.1. Immunofluorescence Staining

Consecutive staining of 5 markers was performed on 3 µm sections of all myocardial samples on the same slide to enable colocalization assessment as described by Klinge et al. [22]. Immunofluorescence staining was carried out using primary antibodies for CD41 (ab203189, Abcam, Cambridge, UK), CD68 (ARP63008-P050; Aviva Systems Biology, San Diego, CA, USA), MPO (ab208670, Abcam, Cambridge, UK), and HIF2a (NB100-122, Novus Biologicals, Wiesbaden, Germany) according to a standard protocol. A secondary antibody, ImmPRESS^®^ HRP Horse Anti-Rabbit IgG Polymer Detection Kit-Peroxidase (MP-7401; Vector Laboratories, Newark, NJ, USA), was used and coupled with an Opal dye, respectively. CD41 was labeled with Opal 480 (P1500001KT, Akoya Biosciences, Menlo Park, CA, USA), HIF2a with Opal 520 (FP1487001KT, Akoya Biosciences, Menlo Park, CA, USA), CD68 with Opal 570 (FP1488001KT, Akoya Biosciences, Menlo Park, CA, USA), and MPO with Opal 650 (FP1496001KT, Akoya Biosciences, Menlo Park, CA, USA). Hypoxyprobe™ was included as the fifth marker in this panel, as described below.

#### 2.5.2. Hypoxyprobe™

The central venous catheter was used to administer an intravenous infusion of Hypoxyprobe™ TM-1 solution (pimonidazole HCl) at a dose of 60 mg/kg BW 60 min before sacrifice. Pimonidazole distributes to all tissues after injection and binds to thiol-containing proteins only in cells with an oxygen partial pressure < 10 mmHg. Subsequently, immunofluorescence staining was performed as described above using a mouse anti-pimonidazole, FITC-conjugated IgG1 monoclonal antibody from the Hypoxyprobe™ Kit (HP2-1000 Kit, Hypoxyprobe™, Inc. Burlington, MA, USA) as the primary antibody. The secondary antibody was then conjugated to Opal 780 (FP1501001KT, Akoya Biosciences, Menlo Park, CA, USA).

#### 2.5.3. Fluorescence Imaging

Fluorescence imaging was conducted using an Axio Imager 2 microscope (20×, ZEISS, Jena, Germany) equipped with a Colibri 7 light source (ZEISS, Germany) and the TissueFAXS PLUS system (TissueGnostics, Vienna, Austria). The light source comprises six LED modules and seven fluorescence channels, each emitting monochromatic light of a different wavelength. LED-optimized filters and direct coupling enhance sensitivity and ensure optimal excitation and emission spectra.

The images were analyzed using StrataQuest Analysis Software (v7, TissueGnostics, Vienna, Austria), and the minimum and maximum ranges for each filter were set by automatically adjusting the saturation. The mean minimum and maximum intensities of the slides were used for each marker in each panel [22].

Nuclei were detected and segmented using DAPI images. The mean staining intensities for the five different markers were measured using nuclei areas in six equally distributed circular ROIs with an area of 0.785 mm^2^ (a circular ROI that fits into a 1 × 1 cm square) per slide. The total number of cells with a mean intensity greater than 100, which were considered as ‘positive’, was recorded.

### 2.6. Blood and Serum Analysis

To analyze the BDL-induced liver injury, alkaline phosphatase (AP), alanine aminotransferase (ALT), aspartate aminotransferase (AST), gamma-glutamyl transpeptidase (GGT), and total bilirubin were determined (Laboratory of Hematology at the Institute of Laboratory Animal Science and Experimental Surgery, RWTH Aachen University, Faculty of Medicine, Aachen, Germany). To determine a potential myocardial injury creatine kinase (CK), creatine kinase MB isoenzyme (CKMB), cardiac troponin I (cTnI; Rat Cardiac Troponin-I ELISA, CTNI-2-HS; Life Diagnostics, Inc., West Chester, PA, USA), and serum brain natriuretic peptide (BNP; Rat BNP 45 ELISA Kit ab108816; Bertha-Benz-Straße 5, 10557 Berlin, Germany) were analyzed. White blood cell (WBC) counts, platelets (PLT), and hemoglobin (Hb) were measured using a cell counter (Celltac Alpha VET MEK-6550, Nihon Kohden, Tokyo, Japan) and lactate was measured using an ABL800 FLEX blood gas analyzer (Radiometer GmbH, Europark Fichtenhain A 4, 47807 Krefeld, Germany). Tumor necrosis factor α (TNFα; Rat TNF-alpha Quantikine ELISA Kit, RTA00; R&D Systems, Inc. Minneapolis, MN, USA) was measured using a commercially available ELISA kit. The total amount of circulating bile acids was determined using a Total bile acids assay (Total Bile Acids (TBA) Assay Kit MAK382, Merck, Sigma-Aldrich Chemie GmbH, Eschenstr. 5, 82024 Taufkirchen, Germany). All assays were performed according to the respective manufacturer’s protocols.

### 2.7. Statistics

Statistical analysis and presentation of data were performed using GraphPad Prism 9 (version 9.2.0, GraphPad Software, Inc., La Jolla, CA, USA) and JMP Pro 15.2.1 (SAS Institute, Cary, NC, USA) and SAS software 9.4 (SAS Institute, Cary, NC, USA).

Data are presented as mean ± standard deviation (SD). Normality and homoscedasticity of data were checked by diagnostic plots (residuals and quantile plots). Overall testing was performed using a generalized linear mixed model analysis considering group and time for serum parameters and group, time, and the effect of pharmacologically induced stress for hemodynamic measurements. In the case of heteroscedasticity, the heterogenous variances were accounted for in the model. All *p*-values from parametric analysis were adjusted for multiple comparisons using the Shaffer-simulated approach.

Where the distribution of the data was unknown, a nonparametric test was chosen (e.g., Kruskall–Wallis) and the resulting *p*-values were adjusted for multiple comparisons using FDR (False Discovery Rate) [23]. The null hypothesis was rejected for *p* < 0.05.

## 3. Results

A total of 37 animals underwent surgery, of which 5 died (4 BDL and 1 control). Two animals died due to surgical complications after BDL while two others died during central venous catheterization 5 days post-BDL. One animal reached drop-out criteria 2 days after BDL and had to be withdrawn from the study before completion.

### 3.1. Serum Parameters

The total bile acids assay showed massively increased levels of total circulating bile acids 5 days after BDL compared to controls (Table 1, control: 0.81 ± 2.54 µmol/L vs. BDL: 127.52 ± 57.03 µmol/L; *p* < 0.001). In line with this, BDL animals had massively elevated bilirubin levels (Table 1, *p* < 0.001). 

Serum analysis revealed significantly increased levels of alkaline phosphatase (AP), aspartate transaminase (AST), and gamma-glutamyl transferase (GGT) 5 days after BDL compared to the control group (Table 1, *p* < 0.001). The alanine transaminase (ALT) levels were twice as high in BDL animals on average, but not statistically significant (Table 1). Serum albumin levels were slightly higher in the BDL group compared to the control group (Table 1). Creatinine was unaffected in both groups (Table 1, control: 48.33 ± 5.64 µmol/L vs. 60.91 ± 35.97 µmol/L, *p* = 0.48) 

B-type natriuretic peptide (BNP) levels showed a non-significant trend with higher values on average in the BDL group (Table 1, control: 218.02 ± 207.96 pg/mL vs. BDL: 335.66 ± 557.62 pg/mL, *p* = 0.21). However, creatine kinase (CK) and its MB isoform (CKMB) showed no difference (Table 1). Troponin I also was not significantly different between the groups (Table 1).

Tumor necrosis factor-α (TNF-α) levels were elevated (Table 1, control: 8.16 ± 0.35 pg/mL vs. BDL: 8.54 ± 0.47 pg/mL *p* = 0.047), but without a concomitant increase in white blood cell count.

### 3.2. Transthoracic Echocardiography

#### 3.2.1. Myocardial Function

In accordance with the study design, no significant differences were found on day 0 between animals of both groups that entered the experiments (Appendix A). 

During stress echocardiography on day 5, all tested parameters reflected a significant effect of pharmacological stress (P_stress_ < 0.001). Chronotropic incompetence was seen in rats 5 days after BDL, as controls showed a significantly greater increase in heart rate (HR) in response to pharmacological stress, both to adenosine and to dobutamine (Figure 1a; adenosine control: 320 ± 32/min vs. BDL 291 ± 25/min, *p* = 0.006; dobutamine control: 405 ± 13/min vs. BDL 378 ± 28/min, *p* = 0.021, P_group_ = 0.004, P_stress_ < 0.001, P_group*stress_ = 0.49). 

At rest, a reduced cardiac output (CO) and global longitudinal strain (GLS) were observed 5 days after BDL compared to sham-operated controls (CO: *p* = 0.041; GLS: *p* = 0.01; Figure 1). The ejection fraction (EF), stroke volume (SV), CO, and GLS were impaired during adenosine stress (EF: *p* = 0.011, SV: *p* = 0.005, CO: *p* = 0.002, GLS: *p* = 0.008; Figure 1). During dobutamine stress CO and GLS were significantly lower after BDL (CO: *p* = 0.017, GLS: *p* = 0.002; Figure 1). End-diastolic volume (VED) remained unaffected by BDL (*p* = 0.15; Figure 1).

#### 3.2.2. Blood Pressure

Five days after BDL, animals showed no significant differences in systolic blood pressure or rate pressure product at baseline or during pharmacological stress testing (Figure 2).

#### 3.2.3. Myocardial Contrast Echocardiography and Pulsed-Wave Doppler

Investigation of myocardial perfusion by MCE showed a reduced normalized wash-in rate (WIR) at rest in the BDL animals compared to controls (*p* = 0.03) while no significant group differences were seen in the normalized peak enhancement (PE) or normalized area under the curve (AUC), either at rest or under pharmacologic stress (Figure 3). CFVR determined by Doppler examination of the LAD, was decreased 5 days after BDL under dobutamine compared to controls but showed no difference under adenosine (CFVR P_dobutamine_ = 0.047, P_adenosine_ = 0.73; Figure 4). The diastolic peak velocity (DPV) was also decreased under dobutamine stress (P_dobutamine_ = 0.009; Figure 4), whereas the velocity time integral (VTI) showed no group-specific differences.

### 3.3. Immunohistology

Hypoxyprobe™ (pimondazole) was utilized to investigate whether transient tissue hypoxia had occurred. Significantly more cells were stained with pimonidazole in the myocardium of BDL animals compared to the respective controls (*p* = 0.047, Figure 5—Hypoxyprobe™). No difference was seen for hypoxia-induced factor 2α (HIF2α) positive cells (Figure 5—HIF2α).

Consistent with the increased TNFα serum levels (Table 1), a pronounced neutrophil infiltration of the myocardium was observed 5 days after BDL (*p* = 0.014; Figure 5—MPO) but without a significantly higher number of tissue macrophages (Figure 5—CD68). Moreover, there were no observed differences in the occurrence of microthrombi (Figure 5—CD41).

## 4. Discussion

The present study characterizes the early effects of BDL-induced acute liver injury on cardiac function, perfusion, and structure in rats. Five days following ligation of the common bile duct, the animals developed signs of acute liver injury while at the same time, there was a marked reduction in cardiac function at rest and a reduced contractile reserve in the BDL animals. This was accompanied by tissue hypoxia in the myocardium with significant neutrophil infiltration, yet without signs of irreversible myocardial injury. This might be mediated by CMD as echocardiography showed both reduced myocardial blood flow at rest as well as a reduced CFVR.

Echocardiography revealed a notable decrease in cardiac function 5 days after BDL (Figure 1). Even at rest, reduced contractility (Figure 1f) was observed, which led to a decrease in CO in the animals experiencing acute cholestasis-induced liver injury. This effect worsened during pharmacological stress induced either by adenosine or dobutamine (Figure 1). Stress echocardiography further demonstrated that BDL animals had a reduced ability to increase heart rate in response to dobutamine, indicating chronotropic incompetence, which is also a known hallmark of cirrhotic cardiomyopathy (CCM) [24]. In CCM, chronotropic incompetence is presumably mediated by a diminished β_1_ receptor density in response to a chronically higher level of noradrenaline [25]. In contrast, chronotropic incompetence in acute cholestatic liver failure appears to be caused by high levels of circulating bile acids and direct bile acid signaling, which tend to reduce receptor density but also receptor affinity [26,27,28] (Table 1—TBA). Additionally, a blunted heart rate response to adenosine, a known predictor of cardiac death [29], was observed in the BDL animals. In line with this, both EF and SV were reduced under adenosine stress (Figure 1b,c). Since the VED showed no difference here, this effect seemed to be preload-independent (Figure 1d). 

The impaired contractility and stress response was paralleled by CMD as could be seen in the MCE showing a reduced perfusion at rest, which was also recognizable under stress, at least as a trend (Figure 3). This observation was supported by the significant reduction in CFVR measured by LAD Doppler measurements under dobutamine stress (Figure 4). 

Further investigation using Immunofluorescent staining unveiled myocardial tissue hypoxia (tissue oxygen partial pressure < 10 mmHg; Figure 5 Hypoxyprobe™). However, the temporal relationship between the observed inflammation and hypoxia remains unclear as it is difficult to determine whether the inflammation preceded or was caused by the hypoxia [30]. Even though it is quite conceivable that the inflammation occurred as a consequence of hypoxia, it is well known that inflammatory processes are a key factor in the development of CMD [31]. Notably, serum markers of myocardial injury (CK, CKMB, and Trop I—Table 1) remained unaffected in our experimental setting, except for BNP, which was higher on average in BDL animals, although not significantly (Table 1). The elevated BNP levels were most likely attributed to tissue hypoxia [32].

As there was no evidence of either a coronary obstructive heart disease—more specifically, microvascular occlusion (Figure 5—CD 41)—or an irreversible myocardial injury, a functional disorder can be assumed. This was confirmed through myocardial perfusion measurements and Doppler measurements of the LAD, revealing CMD as already mentioned above. Notably, CFVR—a widely accepted parameter of CMD—was only reduced under dobutamine but not adenosine [31]. This may be due to differences in the mechanisms of myocardial perfusion enhancement under dobutamine and adenosine [33,34]. High bile acid concentrations, as found in the study on hand, were identified as potential modulators reducing the density and affinity of cardiac β-adrenoreceptors, which might explain the reduced CVFR under dobutamine [26]. Whereas coronary vasomotion under adenosine is mediated by different types of adenosine receptors [33] and seemed not to be affected by BDL.

The rate pressure product (RPP) has been used as a surrogate parameter for myocardial oxygen demand [35,36]. However, we did not observe any difference in RPP or in the systolic blood pressure between control and BDL animals (Figure 2), although Doppler and contrast echocardiography measurements revealed a significant difference in perfusion. Thus, the reduction in cardiac function is most likely due to an imbalance between oxygen demand and supply mediated by CMD, which, in turn, could be caused by endothelial dysfunction (ED). Inadequate endothelial nitric oxide (NO) availability, possibly due to inflammatory cytokines, such as TNFα, or the scavenging of vascular NO by free radicals may contribute to ED [37,38]. Ljubuncic et al. consistently demonstrated increased oxidative stress in the myocardium during cholestasis [39]. Elevated TNFα levels in our study (Table 1) further support these findings. As a pro-inflammatory cytokine, TNFα reduces endothelial NO synthetase (eNOS) phosphorylation by increasing oxidative stress (ROS) and eventually inhibiting NO production [38,40], reducing NO-mediated effects such as vasodilation and the modulation of platelet aggregation, as well as leukocyte and monocyte adhesion, may increase vasoconstriction, thrombosis, and endothelial inflammation [41,42]. This may be the reason for CMD, as well as the consecutively elevated myocardial neutrophil infiltration and tissue hypoxia (Figure 5), which, in turn, resulted in reduced cardiac function (Figure 1).

The impairment of cardiac function 5 days after BDL is also consistent with the depletion of cardiac glycogen stores on day 5 described by Tajuddin et al. [43]. Accordingly, the underlying mechanism may also involve a disturbance in energy homeostasis. It is also interesting to note that cardiac glycogen stores return to baseline levels on day 10 after BDL. Unfortunately, we did not record further time points, but it is conceivable that the deterioration in cardiac function observed in our study may be reversed by day 10, and, therefore, day 10 and day 15 should also be recorded in a follow-up project.

Given the absence of a gallbladder in rats, CBD obstruction prompts a rapid 100-fold increase in circulating bile acids (Table 1—TBA). As bile acids are known to have negative chronotropic and potentially cardiotoxic effects on the myocardium, a direct bile acid signaling pathway must be taken into account [27,28,44]. There are at least two known receptors to which bile acids can bind: farnesoid X receptor (FXR) and Takeda G protein-coupled receptor 5 (TGR5), both of which have been shown to be expressed on cardiomyocytes (FXR and TGR5) and in the vasculature (FXR) [45,46]. However, the literature on the effect of receptor agonism on one of the two receptors is somewhat controversial. While Pu et al. demonstrated a proapoptotic effect of FXR agonism via activation of caspases 9 and 3 and the opening of the mitochondrial permeability transition pore [47], several other studies have shown rather cardioprotective effects of receptor activation [48,49,50,51]. FXR activation appears to play an integral role in maintaining the balance of cellular redox status, and its activation exerts antioxidative effects via nuclear factor erythroid 2-related factor-2 [52]. Bayat et al., on the other hand, reported a significant reduction in cardiac FXR expression four weeks after BDL [53]. This suggests that the FXR-dependent antioxidant effects may be limited in cholestasis, allowing ROS-mediated vasoconstriction to prevail, affecting the observed deficit in myocardial perfusion. Moreover, the effects of bile acids on arrhythmias appear to be dependent on the concentration and composition of total circulating bile acids. Patients with arrhythmias have lower serum levels of hydrophilic bile acid ursodeoxycholic acid (UDCA) and higher serum levels of non-UDCA bile acids [27]. The chronotropic incompetence in the BDL animals we saw might therefore be attributed to the high levels of circulating bile acids. 

The lack of a breakdown of total circulating bile acids into their individual components is certainly a limitation of our study. Nevertheless, the bile acid composition in BDL experiments in rats has already been discussed elsewhere [54,55,56]. 

Furthermore, it is noteworthy that the HIF-2a data did not correlate with the Hypoxyprobe™ data, although HIF-2a showed a trend toward BDL. However, prior studies have already demonstrated that there is often a lack of colocalization and correlation between Hypoxyprobe™ and other hypoxia markers [57]. This divergence likely arises because Hypoxyprobe™ selectively binds to hypoxic tissue [58,59], while other hypoxia markers such as HIF-1a, HIF-2a, or VEGF for instance are subject to multifactorial regulatory mechanisms.

When comparing our data with other studies, it should be noted that the animals have a relatively high weight of 500 ± 23 g. However, the animals are still in the adolescent stage before the age of 8 weeks and are therefore not yet fully grown [60]. As acquired liver diseases tend to affect adults, we have chosen to use 12-week-old adult animals weighing approximately 500 g. In addition, there should be no difference in the effectiveness of the BDL in heavier animals.

As we observed a dropout rate of 20% in a previous study investigating CCM, partly with macroscopically visible infarcts, we formed the hypothesis that inflammation, endothelial dysfunction, and direct bile acid signaling might be involved in the pathogenesis of secondary myocardial injury in ALF [11]. Inflammatory infiltration of the myocardium and tissue hypoxia was verified and is indeed likely to be related to endothelial dysfunction. Additionally, high levels of circulating bile acids are concomitant at least, making direct bile acid signaling very likely (Figure 5—MPO and Hypoxyprobe, Table 1—TBA). However, part of our hypothesis was also that there is an increased formation of microthrombi. Our current study did not confirm this, as CD41 staining revealed no difference between groups (Figure 5—CD41).

## 5. Conclusions

In summary, acute cholestasis-induced liver injury caused myocardial depression in rats due to coronary microvascular dysfunction most likely affected by an endothelial dysfunction. Further investigation is warranted to elucidate the exact cellular and molecular mechanisms mediating the coronary microvascular dysfunction. 

## Figures and Tables

**Figure 1 biomedicines-12-00876-f001:**
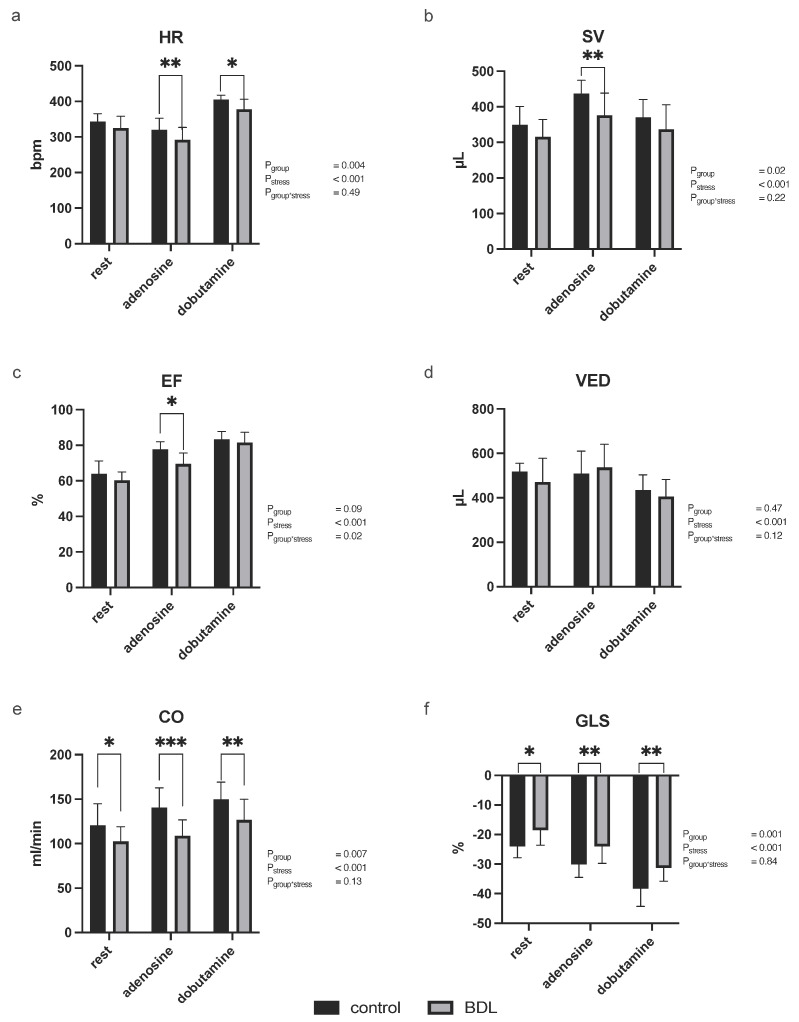
Transthoracic echocardiography in rats showed cardiac dysfunction and an impaired stress response 5 d after bile duct ligation (BDL). Five days post −BDL, *Sprague-Dawley* rats showed chronotropic incompetence (**a**), a reduced stroke volume during adenosine stress (**b**), as well as a reduced ejection fraction (EF; (**c**)), while the end-diastolic volume (VED) was unaffected by BDL (**d**). Cardiac output was reduced at rest as well as under adenosine and dobutamine stress (**e**). In addition to that, a reduced global longitudinal strain (GLS) could be seen (**f**). All parameters reflected a sufficient effect of pharmacological stress (P_stress_ < 0.001). Data are shown as mean ± SD. *p* < 0.05 was considered significant (* *p* < 0.05, ** *p* < 0.01, *** *p* < 0.001).

**Figure 2 biomedicines-12-00876-f002:**
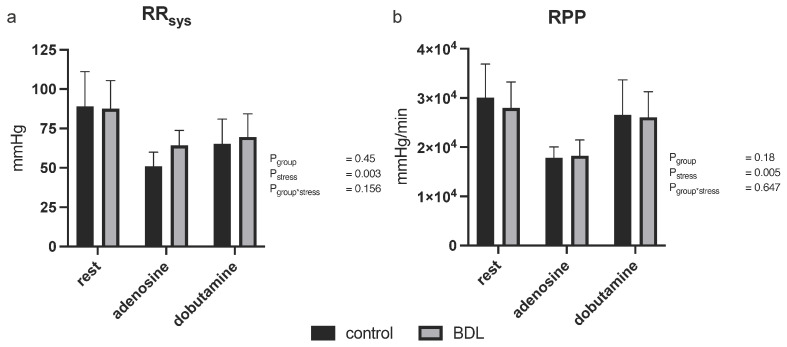
Systolic blood pressure (RR_sys_) and rate pressure product (RPP) in rats 5 days after bile duct ligation (BDL). A possible effect of blood pressure on myocardial perfusion in the BDL rats could be excluded, as RR_sys_ (**a**) and RPP (**b**) showed no significant difference between the groups. All tested parameters reflected a sufficient effect of pharmacological stress (P_stress_ < 0.01). Data are shown as mean ± SD. *p* < 0.05 was considered significant.

**Figure 3 biomedicines-12-00876-f003:**
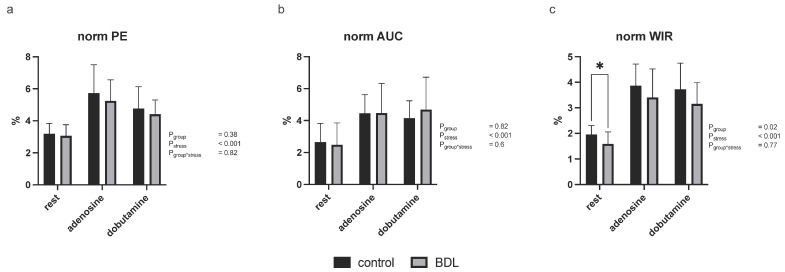
Myocardial contrast echocardiography derived perfusion parameters in rats 5 days after bile duct ligation (BDL). Myocardial contrast echocardiography revealed a significantly reduced normalized wash-in rate (WIR; (**c**)) at rest. Normalized peak enhancement (PE; (**a**)) showed a trend toward less perfusion in the BDL animals, while normalized area under the curve (AUC; (**b**)) did not reveal any difference. All tested parameters reflected a sufficient effect of pharmacological stress (P_stress_ < 0.001). Data are shown as mean ± SD. *p* < 0.05 was considered significant (* *p* < 0.05).

**Figure 4 biomedicines-12-00876-f004:**
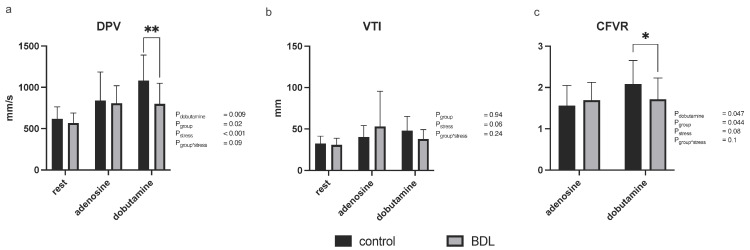
Coronary artery Pulsed-wave Doppler measurements in rats 5 days after bile duct ligation (BDL). Coronary Doppler measurements showed a decreased diastolic peak velocity (DPV; (**a**)) as well as a reduced coronary flow velocity reserve (CFVR; (**c**)) under dobutamine stress. The velocity time integral (VTI; (**b**)) showed no group differences. Data are shown as mean ± SD. *p* < 0.05 was considered significant (* *p* < 0.05, ** *p* < 0.01).

**Figure 5 biomedicines-12-00876-f005:**
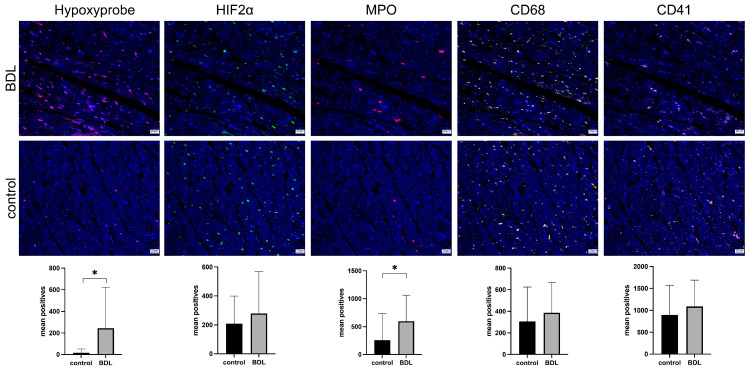
Rats showed signs of myocardial hypoxia and neutrophil infiltration 5 days after bile duct ligation (BDL). Five days after BDL significantly more markers of tissue hypoxia could be seen in rat myocardium compared to controls (hypoxyprobe, *p* = 0.047). In line with that, hypoxia-induced factor 2α (HIF2α) shows a trend toward BDL. Furthermore, a neutrophil infiltration could be seen in BDL rats (MPO, *p* = 0.014). No differences were seen for macrophage infiltration (CD68) or microthrombi (CD41). For each marker, six equally sized regions of interest per slide were automatically counted using Strataquest. Fluorescence microscopic images are shown, and the scalebar corresponds to 20 µm. Data are shown as mean positive counts per slide ± SD. *p* < 0.05 was considered significant (* *p* < 0.05).

**Table 1 biomedicines-12-00876-t001:** Serum parameters of liver dysfunction, myocardial injury, and inflammation.

Parameter	Control (Mean ± SD)	BDL (Mean ± SD)	*p*-Value	
alkaline phosphatase (U/L) *n*	166.78 ± 36.82	275.31 ± 66.58	<0.001	***
	9	23		
aspartate transaminase (U/L) *n*	55.33 ± 6.87	338.61 ± 204.68	<0.001	***
	9	23		
alanine transaminase (U/L) *n*	95.33 ± 86.46	160.65 ± 109.16	0.06	ns
	9	23		
gamma-glutamyl transferase (U/L) *n*	10.38 ± 2.56	35.48 ± 10.91	<0.001	***
	8	23		
albumin (g/dL) *n*	4.14 ± 0.4	4.67 ± 0.66	0.013	*
	8	23		
creatinine (µmol/L) *n*	48.33 ± 5.64	60.91 ± 35.97	0.48	ns
	9	21		
total bile acids (µmol/L) *n*	0.81 ± 2.54	127.52 ± 57.03	<0.001	***
	9	22		
bilirubin (µmol/L) *n*	2 ± 0	169.44 ± 44.55	<0.001	***
	6	23		
creatine kinase (U/L) *n*	92.56 ± 36.16	116.52 ± 66.05	0.31	ns
	9	23		
creatine kinase MB isoenzyme (U/L) *n*	178.22 ± 87.82	139.83 ± 72.35	0.21	ns
	9	23		
troponin I (pg/mL) *n*	15.66 ± 13.69	7.64 ± 9.06	0.07	ns
	9	20		
brain natriuretic peptide (pg/mL) *n*	218.02 ± 207.96	335.66 ± 557.62	0.21	ns
	4	17		
white blood cells (10^3^/µL) *n*	11.11 ± 3.86	11.57 ± 4.62	0.81	ns
	8	21		
tumor necrosis factor-α (pg/mL) *n*	8.16 ± 0.35	8.54 ± 0.47	0.047	*
	8	25		
platelets (10^3^/µL) *n*	305.4 ± 180.71	407.8 ± 202.82	0.16	ns
	7	20		
hemoglobin (g/dL) *n*	13.87 ± 0.62	13.12 ± 1.98	0.2	ns
	6	22		
lactate (mmol/L) *n*	3.31 ± 2.3	2.81 ± 1.55	0.36	ns
	8	23		

Serum markers of liver dysfunction, myocardial injury, and inflammation were obtained in bile duct ligated (BDL) rats 5 days post BDL. Data were analyzed using a two-sided *t*-test where appropriate or a non-parametric test. *p* < 0.05 was considered significant (* *p* < 0.05, *** *p* < 0.001, ns = not significant). The data are presented as mean ± standard deviation (SD).

## Data Availability

The raw data of the laboratory values generated and analyzed during the current study are available from the corresponding author on reasonable request.

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
