# Peer review of "Coronary Microvascular Dysfunction in Acute Cholestasis-Induced Liver Injury"

_biomedicines, 2024, doi:10.3390/biomedicines12040876_

Round 1

Reviewer 1 Report

Comments and Suggestions for Authors

Major revision

The current manuscript reported the acute cholestatic liver failure can cause coronary microvascular dysfunction in animals due to myocardial hypoxia and neutrophil infiltration. The logistics of this manuscript is good, and the method used can refer to the molecular mechanisms of BDL-induced heart failure. My concerns are listed as followed:

1.In the abstract, they have to indicate the possible pathways of coronary microvascular dysfunction caused by acute cholestatic liver failure as revealed by immunofluorescence staining.

2.The body weight of the rats used is around 500g. The body weight is too high. They have to discuss why they used too heavy rats.

3.It is better that they provide the HE staining for the heart to show the change of heart micro-structure after BDL. 

Reviewer 2 Report

Comments and Suggestions for Authors

My comments

1. Liver failure is diagnosed with a combination of INR >1.5 plus hepatic encepahlopathy. The data on these ALF defining parameters are missing. Further, marked ALT elevation is a strong feature of ALF which is not the case here. If we do not have ALF defining data, then we shall use the term 'cholestasis induced liver injury' instead of 'cholestasis induced liver failure' all over the manuscript, from title to conclusion.

2. Data shall be expressed and compared as median instead of mean

3. Please give IQR instead of SEM. Use of SEM under-represent the data variability. 

Comments on the Quality of English Language

It is ok

Reviewer 3 Report

Comments and Suggestions for Authors

Billig S. et al. conducted an experimental study using a rodent bile duct ligation (BDL) model to investigate the short-term effects of acute cholestatic liver failure on cardiac function and structure. They found that BDL animals exhibited acute liver injury with elevated transaminases, bilirubin, and total circulating bile acids. Concurrently, cardiac function was significantly impaired, characterized by reduced cardiac output and global longitudinal strain. Myocardial perfusion analysis revealed reduced blood flow and coronary flow velocity reserve in BDL animals. Immunofluorescence staining indicated myocardial hypoxia and increased neutrophil infiltration, suggesting that acute cholestatic liver failure can impair cardiac function mediated by coronary microvascular dysfunction, potentially contributing to mortality in acute liver failure.

1.     The introduction provides a solid foundation for the study; however, further elucidation of the molecular pathways involved in the association between acute cholestatic liver failure and cardiac function could significantly enhance readers' understanding. Incorporating discussions on key molecular pathways such as inflammatory mediators, oxidative stress, and hormonal dysregulation would deepen the context and relevance of the study.

2.     While the aim of the study is implied, explicitly stating it in the introduction would provide clarity for readers. Consider revising to: "In this study, our aim is to investigate the short-term effects of acute cholestatic liver failure on cardiac function and structure using a rodent model."

3.     The choice of the rodent bile duct ligation (BDL) model as a representative model for acute cholestatic liver failure is crucial. Providing a brief explanation in the introduction on why this model was selected and how it mimics clinical scenarios would strengthen the rationale behind the experimental approach.

4.     Concluding the introduction with a clear statement of the research hypothesis or objective would effectively guide readers through the study's purpose. Emphasize the importance of understanding cardiac complications in acute liver failure and how this study contributes to addressing this knowledge gap.

5.     Consider discussing the rationale behind choosing a short-term assessment (5 days) in contrast to long-term outcomes to enhance the clinical relevance of the findings. Referring to previous literature, such as reference (1), which suggests the reversal of bile acid and other parameters towards normal levels by the 10th and 15th day post-BDL, could provide insight into the temporal dynamics of coronary microvascular dysfunction and its implications for long-term cardiac function and mortality outcomes.

6.     Incorporating measurements of renin-angiotensin system (RAS) components would provide valuable insights into the pathophysiological mechanisms underlying cardiac dysfunction in acute liver failure. Consider including this analysis to complement the study's findings and broaden its scope.

References:

11. Tajuddin M, Tariq M, Bilgrami NL, Kumar S. Biochemical and pathological changes in the heart following bile duct ligation. Adv Myocardiol. 1980;2:209-12. PMID: 7423038.

Round 2

Reviewer 2 Report

Comments and Suggestions for Authors

Dear Authors

Thanks for accepting my suggestions